# VC DIMENSIONS FOR DEEP NEURAL NETWORKS WITH BOUNDED-RANK WEIGHT MATRICES

## ABSTRACT

Deep neural networks (DNNs) have seen immense success in the past decade, yet their lack of interpretability remains a challenge. Recent research on the VC (Vapnik-Chervonenkis) dimension of DNNs has provided valuable insights into the underlying mechanisms of deep learning's powerful generalization capabilities. Understanding the VC dimension offers a promising path toward unraveling the enigma of deep learning, ultimately leading to more interpretable and trustworthy AI systems. In this paper, we study the VC dimensions for DNNs with piecewise polynomial activations and bounded-rank weight matrices. Our main results show that the VC dimensions for DNNs with weight matrices that have bounded rank $r$ are at most $\mathcal{O}(nrL^2 \log(nrL))$, where $n$ is the width of the network, and $L$ is the depth of the network. We also construct a ReLU DNN with bounded rank $r$ that can achieve the VC dimension $\Omega(nr)$, which confirms that the upper bound we obtain is nearly tight for large $n$. Based on these bounds, we compare the generalization power in terms of VC dimensions for various different DNN architectures.

## 1 INTRODUCTION

In the past decade, deep neural networks (DNNs) have achieved remarkable success across a wide range of applications, such as image classification, natural language processing, and autonomous driving (see Hinton et al., 2012; Goodfellow et al., 2013; Abdel-Hamid et al., 2014; Silver et al., 2016; Vaswani et al., 2017; Devlin et al., 2018; Dosovitskiy et al., 2020). These deep learning models, inspired by the structure of the human brain, have demonstrated an unprecedented capacity to learn complex patterns and representations from vast amounts of data. This has led to breakthroughs in fields such as computer vision, speech recognition, and recommendation systems, revolutionizing industries and reshaping our daily lives.

One of the most impressive aspects of deep neural networks is their ability to generalize from the data they have been trained on to make accurate predictions on unseen examples (White, 1992; Anthony et al., 1999; Goodfellow et al., 2016; Allen-Zhu et al., 2019). This phenomenon, known as generalization, is at the core of the deep learning success story. Despite the staggering performance of these models, there remains a significant challenge in understanding how and why they work so effectively in practical tasks. This challenge is rooted in the lack of interpretability and transparency of deep neural networks. The black-box nature of deep learning models is a well-acknowledged issue. While we can feed them data and obtain predictions, it is often difficult to discern why a particular decision was made. This lack of transparency has raised concerns, especially in applications where interpretability is crucial, such as healthcare and autonomous vehicles (Alzubi et al., 2018; Bashar, 2019; Azghadi et al., 2020). Researchers and practitioners have been diligently working to unravel the mysteries of deep learning, striving to make these models more interpretable and transparent.

One avenue of research that has gained prominence in recent years is the study of VC (Vapnik-Chervonenkis) dimension (Vapnik, 1968) and its relationship with the generalization capabilities of deep neural networks (Sontag et al., 1998; Anthony et al., 1999; Goodfellow et al., 2016; Allen-Zhu et al., 2019). VC dimension is a concept from statistical learning theory that measures the capacity of a model to fit a wide variety of datasets while avoiding overfitting. It provides insights into the trade-off between model complexity and generalization performance. Understanding the VC dimension

of deep learning architectures can shed light on why these models generalize so well despite their complexity (see Baum & Haussler, 1988; Karpinski & Macintyre, 1997; Vapnik & Chervonenkis, 2015; Anthony & Bartlett, 2009; Bartlett et al., 1998; 2019; Wang & Scott, 2021).

The study of VC dimensions of DNNs goes back to the 1980s. For instance, In 1989, Baum and Haussler derived a bound for the VC dimension for linear threshold neural networks (Baum & Haussler, 1988). Later in 1997, several polynomial bounds for VC dimensions of sigmoidal and general Pfaffian NNs were given in (Karpinski & Macintyre, 1997). In 1998, Sontag established various elementary results of the VC dimensions of some simple networks in (Sontag et al., 1998). He found out that the VC dimensions of the perceptrons and single hidden layer networks with fixed input weights and activation function $\mathtt{tanh}$ are exactly equal to the number of parameters in the networks. Finally, they also derived a tight upper bound of the class of functions computed by DNNs with the Heaviside activation function. Recently, Bartlett and Harvey studied the VC dimension of DNNs with piecewise polynomial activation functions in (Bartlett et al., 2019) and derived several sharp lower and upper bounds. The network studied in (Sontag et al., 1998) is a special case of that in (Bartlett et al., 2019) since the Heaviside function is a special case of the piecewise polynomial with degree equal to 0.The VC dimension has also been extensively studied in various network architectures. In 1997, Koiran and Sontag studied the order of growth of the VC dimensions of recurrent neural networks for different activation functions in (Koiran & Sontag, 1997). In 2018, Scarselli, Tsoi and Hagenbuchner studied the upper bound of the VC dimension for the set of graph neural networks for different activation functions (polynomials, piecewise polynomials and tanh) in (Scarselli et al., 2018). However, currently, there are no results on the VC dimension of DNNs with low-rank weight matrices, while a recent result (Galanti et al., 2022) shows that the weight matrices after training are often very close to low-rank weight matrices in practice. This makes it important to study the VC dimension of DNNs with bounded-rank weight matrices, and our paper aims to fill this gap.

**Our Contributions.** In this paper, we study the VC dimension of DNNs with bounded-rank weight matrices and piecewise polynomial activation functions. *To the best of our knowledge, our paper is the first work on calculating the lower and upper bounds of VC dimensions for DNNs with bounded-rank weight matrices.* The main contributions of this work are:

- We derive some upper bounds for the VC dimension of DNNs with bounded-rank weight matrices and piecewise polynomial activation functions.

- Furthermore, by construction, we show that there exists such type of DNNs that can achieve the VC dimension close to the upper bounds we derived, which shows that our upper bound is nearly tight.

- Based on these bounds, we compare the generalization power in terms of VC dimensions for various different DNN architectures.

The rest of this paper is organized as follows. In Section 2, we introduce some important concepts and fix some notations that will be used in this paper. In Sections 3 and 4, we provide some upper and lower bounds for VC dimensions of fully connected neural networks with bounded rank weight matrices. In section 5, we construct a class of DNNs that can achieve the VC dimension close to the upper bounds we obtained, thus verifying that our upper bound is nearly tight. Finally, we provide the conclusion and discuss several future directions in Section 6. Most of the proofs of our results are given in the Appendix.

## 2 PRELIMINARY

In this section, we introduce introduce some concepts and fix some notations.

**Definition 1** (Asymptotic Notations $\mathcal{O}(\cdot), \Omega(\cdot)$ and $\Theta(\cdot)$). *For two functions $f(n)$ and $g(n)$, we write $f(n) = \mathcal{O}(g(n))$ if there exists some positive constant $c > 0$ such that $f(n) \leq cg(n)$ for all $n$ larger than some constant; $f(n) = \Omega(g(n))$ if there exists some positive constant $c$ such that $f(n) \geq cg(n)$ for all $n$ large enough; and $f(n) = \Theta(g(n))$ if there exists some positive constants $c_1, c_2$ such that $c_1 g(n) \leq f(n) \leq c_2 g(n)$ for all $n$ large enough.*

**Definition 2** (Sign Patterns and VC-Dimensions). *(Vapnik, 1968) Let $\mathcal{F}$ denote a class of functions from the input set $X$ to $\{-1, +1\}$. Let $\boldsymbol{x}_1, \ldots, \boldsymbol{x}_m \in \mathcal{F}$. For any non-negative integer $m$ and*

$f \in \mathcal{F}$, a length $m$ sign vector

$$(f(\boldsymbol{x}_1), \ldots, f(\boldsymbol{x}_m))$$

denotes as a sign pattern generated by $f$. We define the number of sign patterns generated by $\mathcal{F}$ on $\{\boldsymbol{x}_1, \ldots, \boldsymbol{x}_m\}$ as

$$\gamma_{\mathcal{F}}(\{\boldsymbol{x}_1, \ldots, \boldsymbol{x}_m\}) := |\{(f(\boldsymbol{x}_1), \ldots, f(\boldsymbol{x}_m)) : f \in \mathcal{F}\}|.$$

We say $\{\boldsymbol{x}_1, \ldots, \boldsymbol{x}_m\}$ is shattered by $\mathcal{F}$ if and only if

$$\gamma_{\mathcal{F}}(\{\boldsymbol{x}_1, \ldots, \boldsymbol{x}_m\}) = 2^m.$$

The VC (Vapnik–Chervonenkis) dimension of $\mathcal{F}$ is defined as

$$VCD(\mathcal{F}) := sup\{m \in \mathbb{N} : \exists (\boldsymbol{x}_1, \ldots, \boldsymbol{x}_m) \in X^m \text{ such that } \{\boldsymbol{x}_1, \ldots, \boldsymbol{x}_m\} \text{ is shattered by } \mathcal{F}\}.$$

**Remark 1.** *Intuitively, $\{x_1, \ldots, x_m\}$ is shattered by $\mathcal{F}$ means that for each length $m$ sign pattern, there must exist a function $f \in \mathcal{F}$ maps this $m$ points to this sign pattern. The largest cardinality of shattered sets of $\mathcal{F}$ is the VC dimension of $\mathcal{F}$. Notice that $VCD(F) = m$ does not mean any arbitrarily $m$ points can be shattered by $F$, it only means that there exist $m$ points that can be shattered by $\mathcal{F}$ and any $n$ points can not be shattered by $\mathcal{F}$ for $n \geq m+1$.*

**Definition 3** (Fully Connected Neural Networks with Bounded-Rank Weight Matrices). *In this paper, we consider a fully connected neural network (FCNN) which can be seen as a function from the input space $\mathcal{X}$ to $\{-1, 1\}$. To be more specific, an FCNN can be structurally defined by a directed acyclic graph $\mathcal{G}$, an activation function $\psi : \mathbb{R} \to \mathbb{R}$ and a set of parameters (weights and biases). Let $L$ be an integer greater than 1. We say that the FCNN has $L$ layers when the acyclic graph has $L+1$ distinct sets of nodes that form $L + 1$ layers of the FCNN. The set $\ell_0$ is called the set of input nodes (input layer) with in-degree 0 and $\ell_L$ is called the set of output nodes (output layer) with out-degree 0. For $1 \leq i \leq L$, all nodes in $\ell_{i-1}$ only connect to all nodes in $\ell_i$ to form edges of $\mathcal{G}$. We use $k_i$ to denote the number of nodes in layer $i$. For example, $k_0$ denotes the dimension of the input vector and $k_L$ denotes the dimension of the output vector. In this paper, we fix $k_L = 1$ for convenience. The weights of this FCNN are a set of real values $\boldsymbol{w} = (w_1, w_2, \ldots, w_W) \in \mathbb{R}^W$ associated with each edge in $\mathcal{G}$. We call all the nodes in $\ell_1, \ldots, \ell_L$ the computational units. let $U$ be the number of computational units and each of them associates with a bias in $\boldsymbol{b} = (b_1, b_2, \ldots, b_U) \in \mathbb{R}^U$ and an activation function. The activation function for the output unit is the sign function defined as*

$$f(x) = \begin{cases} +1, & x > 0 \\ -1, & x \leq 0. \end{cases}$$

*The activation function for other units is $\psi$, which is a given piecewise polynomial function with $p$ pieces and of degree no more than $d$. The computational rule for FCNN proceeds as follows. Let $\boldsymbol{x}_i$ be the output $k_i \times 1$ vector of the layer $i$ and $\boldsymbol{W}_{i+1}$ be the $k_{i+1} \times k_i$ weight matrix and $\boldsymbol{b}_{i+1}$ be a $k_{i+1} \times 1$ bias vector for the input of layer $i + 1$. Then the input of layer $i + 1$ is given by*

$$g_{i+1}(\boldsymbol{x}_i) := \boldsymbol{W}_{i+1}\boldsymbol{x}_i + \boldsymbol{b}_{i+1}.$$

*The output of layer $i + 1$ is given by*

$$\Psi_{i+1}(\boldsymbol{z}) := (\psi(\boldsymbol{z}_1), \ldots, \psi(\boldsymbol{z}_{k_{i+1}}))^T.$$

*For the last layer, the output is equal to*

$$sgn \circ g_{L-1}(\boldsymbol{x}_{L-1}) = sgn(\boldsymbol{W}_L\boldsymbol{x}_{L-1} + \boldsymbol{b}_L).$$

*Hence the function represented by such FCNN can be viewed as the following function from $\mathbb{R}^{k_0}$ to $\{-1, +1\}$:*

$$f_{\boldsymbol{w},\boldsymbol{b}}(\boldsymbol{x}_0) := sgn \circ g_{L-1} \circ \ldots \circ \Psi_2 \circ g_2 \circ \Psi_1 \circ g_1(\boldsymbol{x}_0).$$

*We call a fully connected neural network with bounded-rank weight matrices a BRFCNN. We denote the rank of the weight matrix for the input of layer $i$ as $r_i$ for $1 \leq i \leq L$. Thus $1 \leq r_i \leq min(k_i, k_{i-1})$. In this paper, we use $\mathcal{F}$ to denote the set of BRFCNNs. For example, the set of BRFCNNs with $L$ layers and the rank of all the weight matrices at most $r$ can be defined as:*

$$\{f_{\boldsymbol{w},\boldsymbol{b}}(\cdot) : \boldsymbol{w} \in \mathbb{R}^W, \boldsymbol{b} \in \mathbb{R}^U, r_i \leq r, \forall 1 \leq i \leq L\}.$$

*We use the notation $w_i$ to denote the number of network parameters from layer 1 to layer $i$.*

## 3 UPPER BOUNDS OF VC DIMENSIONS FOR BRFCNNS

In this section, we first recall Lemma 1 in (Bartlett et al., 2019) that was used to find the VC dimension of the upper bound of an FCNN. After that, we will briefly explain why this lemma can not be directly applied to the study of the upper bound for the VC dimension of BRFCNNs. Instead, motivated by Lemma 1, we derive Theorem 1 to adapt the rank constraint of the weight matrices. According to our method, we need to solve two counting problems in order to apply Theorem 1. The first one is the counting problem for the number of free variables in a bounded rank matrix, which will be studied in Section 3.2. The second problem is a degree counting problem of the degrees of polynomials and rational fractions that appear in the functions of BRFCNNs, which will be solved in Section 3.3. After that, we derive the VC upper bounds for BRFCNNs with bounded rank weight matrices and piecewise polynomial activation functions in Theorem 2 and Theorem 3. Finally, in Section 3.5, we compare the VC dimensions for various different DNN architectures.

### 3.1 SOME PRELIMINARY RESULTS FOR STUDYING VC DIMENSIONS OF DNNS

In this subsection, we provide some preliminary results for studying VC Dimensions of DNNs. The first result is from (Bartlett et al., 2019).

**Lemma 1.** *(Bartlett et al., 2019) Let $p_1, \ldots, p_m$ be polynomials of degree at most $d$ in $n \leq m$ variables. Define*

$$K := |\{(sgn(p_1(x)), \ldots, sgn(p_m(x)) : x \in \mathbb{R}^n\}|,$$

*i.e. $K$ is the number of possible sign vectors given by the polynomials. Then $K \leq 2(2emd/n)^n$.*

Lemma 1 is a useful technique to find the upper bounds for the VC dimensions of FCNNs with piecewise polynomial activations. However, this result can not be directly applied to the study of the upper bound for the VC dimension of BRFCNNs with bounded rank weight matrices. Therefore, we modify Lemma 1 to adapt the weight matrices rank constraint, and derive the following theorem that can solve this problem. The proof is given in Appendix A.1.

**Theorem 1.** *Let $f_1, \ldots, f_m$ be rational fractions in $n \leq m$ variables. They are functions of the same variables and the degrees of their denominators and numerators are at most $d_{den}$ and $d_{num}$ respectively. Define*

$$K := |\{(sgn(f_1(x)), \ldots, sgn(f_m(x)) : x \in \mathbb{R}^n\}|,$$

*i.e. $K$ is the number of possible sign vectors given by the rational fractions. Then $K \leq 2(2em(d_{den} + d_{num})/n)^n$.*

### 3.2 FREE VARIABLES COUNTING

In this subsection, we derive the following results, which can solve the counting problem for the number of free variables in a bounded rank matrix.

**Lemma 2.** *Let $M$ be an $n \times m$ matrix with rank $r$. Then it has $(n + m - r) \times r$ free variables.*

Lemma 2 is used as a tool to count the number of free variables(parameters) in BRFCNN. The proof is given in Appendix A.2.

### 3.3 DEGREE COUNTING

In this subsection, we derive the following lemmas, which can be used to count the degrees of polynomials and rational fractions that appear in the functions of BRFCNNs.

**Lemma 3.** *Let $M$ be an $n \times m$ matrix with rank $r$. Then all the non-free variables can be expressed as a ratio of two polynomials of free variables with numerator degree at most $r + 1$ and denominator degree $r$.*

**Remark 2.** *According to the proof in Appendix 3, we know that when replacing all the non-free variables with free variables in BRFCNN, the input for each unit, in each layer and and each network input will become a rational fraction of the free variables. That is the reason why we need to introduce Theorem 1 in Section 3.1.*

**Lemma 4.** *Let $\mathcal{N}$ be an $L$-layer FCNN with piecewise polynomial activation function of degree at most $d$. Let $0 \le i \le L$. The input of any unit in layer $i$ for a fixed network is a polynomial function of network parameters with degree at most $\sum_{t=0}^{i-1} d^t$. (The proof is given in Appendix A.4.)*

**Lemma 5.** *Let $\mathcal{N}$ be an $L$-layer FCNN with piecewise polynomial activation function of degree at most $d$. Let $1 \le i \le l \le L$ and $v_i$ be a weight in layer $i$. Then the input of any unit in layer $l$ for a fixed network input $x_0$ is a polynomial function of $v_i$ of degree no more than $d^{l-i}$. (Proof is given in Appendix A.5)*

**Lemma 6.** *Let $\mathcal{N}$ be an $L$-layer BRFCNN with piecewise polynomial activation function of degree at most $d$. Let $1 \le i, l \le L$, and the weight matrix in layer $i$ has rank $r_i$. Then the input of any unit in layer $l$ is a ratio of two polynomials of network parameters of degrees at most $\sum_{i=1}^{l}(k_{i-1} - r_i) \times (k_i - r_i) \times d^{l-i} \times r_i$ and $\sum_{t=0}^{l-1} d^t + \sum_{i=1}^{l}(k_{i-1} - r_i) \times (k_i - r_i) \times d^{l-i} \times r_i$, respectively. (The proof is given in Appendix A.6).*

### 3.4 Upper Bounds of VC Dimensions of BRFCNNs

In this section, we first introduce a lemma in (Bartlett et al., 2019), which is used for finding the upper bounds of the VC dimensions for DNNs. Then we derive Theorem 2, which gives upper bounds of VC dimensions for BRFCNNs. Finally, in Theorem 3, we give the upper bounds of the VC dimensions for ReLU BRFCNNs.

**Lemma 7.** *(Bartlett et al., 2019) Suppose that $2^m \le 2^t (mr/w)^w$ for some $r \ge 16$ and $m \ge w \ge t \ge 0$. Then, $m \le t + w \log_2(2r \log_2 r)$.*

**Theorem 2.** *Let $\mathcal{F}$ be a BRFCNN with $L > 1$ layers and the piecewise polynomial activation function $\psi$. In addition, $\psi$ has degree at most $d > 1$ and $p \ge 0$ breakpoints $t_1, \ldots, t_p \in \mathbb{R}$. For all $f \in \mathcal{F}$, the weight matrix in layer $i$ has rank $r_i$, and the number of computation units in layer $i$ is $k_i$. Let $w_i$ be the number of network parameters from layer $1$ to layer $i$. Then $VCD(\mathcal{F})$ is at most*

$$\mathcal{O}\left(L + \left(\sum_{l=1}^{L} w_l\right) \log(pR)\right),$$

*where*

$$R := \sum_{l=1}^{L} k_l \left(\sum_{t=0}^{l-1} d^t + 2\sum_{i=1}^{l}(k_{i-1} - r_i)(k_i - r_i)d^{l-i}r_i\right).$$

*Proof.* For input $x \in \mathcal{X}$ and the network parameter vector $a \in \mathbb{R}^W$, let $f(x; \boldsymbol{a})$ denote the input of the last layer of the network. So we can write the function class as

$$\mathcal{F} := \{\operatorname{sgn}(f(x; \boldsymbol{a})) | \boldsymbol{a} \in \mathbb{R}^W\}.$$

Our goal is to determine an upper bound of $VCD(\mathcal{F})$. If we can find the upper bound of the number of signs patterns $\gamma_{\mathcal{F}}(x_1, \ldots, x_m)_{UB}$ generated by $\mathcal{F}$, then we can get an upper bound for $VCD(\mathcal{F})$. Since if $m = VCD(\mathcal{F})$ we have

$$2^m \le \gamma_{\mathcal{F}}(\boldsymbol{x}_1, \ldots, \boldsymbol{x}_m)_{UB}. \tag{1}$$

By solving Inequality (1) with respect to $m$, we can derive an upper bound for $m$. Due to the rank constraint for the weight matrices, $f(x_j; \boldsymbol{a})$ are not necessarily polynomials, instead they could be rational fractions. Hence we can apply Theorem 1 to bound the number of sign patterns for any given $P_\alpha$. After that, by adding up all the number of sign patterns for all regions, we can find an upper bound for $\gamma_{\mathcal{F}}(\boldsymbol{x}_1, \ldots, \boldsymbol{x}_m)$. The partition $S$ is constructed layer by layer, through a sequence $S_0, S_1, S_2, \ldots, S_{L-1}$, which is given below. Let $g_{h,i,\boldsymbol{x}_j,S'}(\boldsymbol{a})$ and $f_{h,i,\boldsymbol{x}_j,S'}(\boldsymbol{a})$ be the input and output of function of unit $h$ in layer $i$ with respect to parameters $\boldsymbol{a}$ vary in region $S'$ for input $\boldsymbol{x}_j$. Let $\boldsymbol{g}_{i,S'}(a)$ and $\boldsymbol{f}_{i,S'}(a)$ be two vector functions of parameters $\boldsymbol{a}$ in region $S'$ which defined as:

$$\boldsymbol{g}_{i,S'}(\boldsymbol{a}) := (g_{1,i,\boldsymbol{x}_1,S'}(\boldsymbol{a}), \ldots, g_{k_i i,\boldsymbol{x}_1,S'}(\boldsymbol{a}), \ldots, g_{k_i,i,\boldsymbol{x}_m,S'}(\boldsymbol{a}))^T;$$

$$\boldsymbol{f}_{i,S'}(\boldsymbol{a}) := (f_{1,i,\boldsymbol{x}_1,S'}(\boldsymbol{a}), \ldots, f_{k_i i,\boldsymbol{x}_1,S'}(\boldsymbol{a}), \ldots, f_{k_i,i,\boldsymbol{x}_m,S'}(\boldsymbol{a}))^T.$$

There are $k_i \times m$ elements for the input and output vectors $\boldsymbol{g}_{i,S'}(\boldsymbol{a})$ and $\boldsymbol{f}_{i,S'}(\boldsymbol{a})$. We set $S_0 = \mathbb{R}^W$, then the function $f_{0,S_0}$ is a constant vector since it is the input of the network. Now suppose $S_0, \ldots, S_{n-1}$ have been defined and each region $P_i \in S_i$ corresponds to a fixed vector function $\boldsymbol{f}_{i,P_i}(\boldsymbol{a})$. We also want $S_n$ to satisfy this property, which also means that $\boldsymbol{f}_{n,P_n}(\boldsymbol{a})$ is a fixed vector function for all $P_n$ in the partition $S_n$. Let $P_{n-1} \in S_{n-1}$ be one of the regions of the partition $S_{n-1}$. By assumption, $\boldsymbol{f}_{n-1,P_{n-1}}(\boldsymbol{a})$ is a fixed vector function and by Definition 3, $\boldsymbol{g}_{n,P_{n-1}}(\boldsymbol{a})$ is also a fixed vector function with respect to $a$. Hence, we can further divide $P_{n-1}$ such that each new region determines which pieces of activation function does $\boldsymbol{g}_{n,P_{n-1}}(\boldsymbol{a})$ in. The $k_n m p \times 1$ sign pattern vectors

$$A_{n,P_{n-1}}(\boldsymbol{a}) := \left(\operatorname{sgn}(\boldsymbol{g}_{n,P_{n-1}}(\boldsymbol{a}) - t_1), \ldots, \operatorname{sgn}(\boldsymbol{g}_{n,P_{n-1}}(\boldsymbol{a}) - t_p)\right)^T$$

can tell us which pieces of $\psi$ do all the $g_{h,n,x_j,P_{n-1}}(a)$ fall in. For example, if there are 5 breakpoints for $\psi$ and we get

$$(g_{h,n,x_j,P_{n-1}}(a) - t_1, \ldots, g_{h,i,x_j,P_{n-1}}(a) - t_5) = (+, +, +, -, -),$$

then we can say $g_{h,n,x_j,P_{n-1}}(a)$ is between breakpoints $t_3$ and $t_4$. Now, dividing $\mathbb{R}^w$ such that each region corresponds to one sign pattern $A_{n,P_{n-1}}(a)$ and intersect these regions with $P_{n-1}$. The intersections will replace $P_{n-1}$. Performing the same operation for all regions of $S_{n-1}$, we get $S_n$ which satisfies the property we need. We also can determine how many sign patterns does $A_{n,P_{n-1}}(a)$ can get for $a$ vary in $P_{n-1}$. By Lemma 6, we can get that each of $g_{h,n,x_j,P_{n-1}}(a) - t$ are rational fractions with denominator and numerator degree at most

$$\sum_{i=1}^{n}(k_{i-1} - r_i) \times (k_i - r_i) \times d^{n-i} \times r_i$$

and

$$\sum_{t=0}^{n-1} d^t + \sum_{i=1}^{n}(k_{i-1} - r_i) \times (k_i - r_i) \times d^{n-i} \times r_i.$$

By Theorem 1, we have

$$\#\{A_{n,P_{n-1}}(a) | a \in P_{n-1}\} \le \Gamma_n :=$$

$$2(2e(k_n m p)(\sum_{t=0}^{n-1} d^t + 2\sum_{i=1}^{n}(k_{i-1} - r_i)(k_i - r_i)d^{n-i}r_i)/w_n)^{w_n}$$

and the number of new regions for the partition $S_n$ is at most

$$\Gamma_n \times Card(S_{n-1}).$$

We also know that the size of $S_0$ is 1, thus the size of $S_{L-1}$ is at most

$$\prod_{l=1}^{L-1}(2(2e(k_l m p)(\sum_{t=0}^{l-1} d^t + 2\sum_{i=1}^{l}(k_{i-1} - r_i)(k_i - r_i)d^{l-i}r_i)/w_l)^{w_l}). \tag{2}$$

Let $P_{L-1} \in S_{L-1}$, the sign patterns for

$$\{(\operatorname{sgn}(f(x_j; a)), \ldots, \operatorname{sgn}(f(x_m; a))) : a \in P_{L-1}\}$$

is at most

$$2(2em(\sum_{t=0}^{L-1} d^t + 2\sum_{i=1}^{L}(k_{i-1} - r_i)(k_i - r_i)d^{L-i}r_i)/w_L)^{w_L}. \tag{3}$$

Therefore $\gamma(x_1, \ldots, x_m)$ is upper bounded by the product of Eq(2) and Eq(3):

$$\gamma_{\mathcal{F}}(x_1, \ldots, x_m) \le \prod_{l=1}^{L}\left(2\left(2ek_l m p\left(\sum_{t=0}^{l-1} d^t + 2\sum_{i=1}^{l}(k_{i-1} - r_i)(k_i - r_i)d^{l-i}r_i\right)/w_l\right)^{w_l}\right).$$

By weighted AM-GM inequality we get

$$\gamma_{\mathcal{F}}(x_1, \ldots, x_m) \leq 2^{L+1} \cdot$$

$$\left( 2emp \sum_{l=1}^{L} k_l \left( \sum_{t=0}^{l-1} d^t + 2 \sum_{i=1}^{l} (k_{i-1} - r_i)(k_i - r_i)d^{l-i}r_i \right) / \sum_{l=1}^{L} w_l \right)^{\sum_{l=1}^{L} w_l}.$$

Similar to (Bartlett et al., 2019), we can define

$$R := \sum_{l=1}^{L} k_l \left( \sum_{t=0}^{l-1} d^t + 2 \sum_{i=1}^{l} (k_{i-1} - r_i)(k_i - r_i)d^{l-i}r_i \right).$$

Then we get

$$\gamma_{\mathcal{F}}(x_1, \ldots, x_m) \leq 2^L \cdot \left( \frac{2empR}{\sum_{l=1}^{L} w_l} \right)^{\sum_{l=1}^{L} w_l}.$$

Let $VCD(\mathcal{F}) = m$. We have

$$2^m \leq 2^L \cdot \left( \frac{2empR}{\sum_{l=1}^{L} w_l} \right)^{\sum_{l=1}^{L} w_l}.$$

Since $L > 1$ implies $2epR \geq 16$, by Lemma 7 we get

$$VCD(\mathcal{F}) \leq L + (\sum_{l=1}^{L} w_l) \log(4epR \log(2epR)) = \mathcal{O}\left( L + (\sum_{l=1}^{L} w_l)(\log_2(p) + \log_2(R)) \right).$$

$\square$

**Theorem 3.** *Let $r, n \in \mathbb{Z}^+$ and $n > r$. Let $\mathcal{F}$ be a ReLU BRFCNN with width at most $n$, depth at most $L$ and weight matrix rank at most $r$, Then the VC dimension for $\mathcal{F}$ is at most*

$$VCD(\mathcal{F}) = \mathcal{O}\left( L^2 nr \log(nrL) \right).$$

*If $n > L$, we have*

$$VCD(\mathcal{F}) = \mathcal{O}\left( L^2 nr \log(n) \right).$$

*If $L > n$, we have*

$$VCD(\mathcal{F}) = \mathcal{O}\left( L^2 nr \log(L) \right).$$

*(The proof is given in Appendix A.7.)*

### 3.5 UPPER BOUND COMPARISON FOR DIFFERENT DNN ARCHITECTURES

In this subsection, we compare the VC dimensions for various different DNN architectures. Let $U$ be the number of nodes from layer 0 to $L - 1$. i.e:

$$U := \sum_{i=0}^{L-1} k_i.$$

First of all, we compare different settings for upper bounds of the VC dimensions of ReLU BRFC-NNs with fixed $U$ and $r$. In addition, the number of units in each layer is the same. We only vary the network width $n$ and the network depth $L$. We prove in Appendix A.8 that for sufficiently large $U$, swapping the dimension of depth and width will increase the upper bounds of the VC dimensions if $n < L$; and decrease the upper bounds of the VC dimensions if $L > n$. Furthermore, the network with $L = U, n = 1$ has the largest VC upper bound, and the network with $L = 2, n = U/2$ has the smallest VC upper bound.

Next, we compare the ReLU BRFCNN with a fixed graph structure but choose different $r$. By Theorem 3, we obtain that the VC upper bound increases linearly as $r$ increases.

Finally, we compare our VC upper bound with (Bartlett et al., 2019). When applying their bound to the set of ReLU FCNNs with width at most $n$, depth at most $L$ and full-rank weight matrices, their bound becomes

$$\mathcal{O}\left( (n^2 L + (L-2)n + 1)L \log(n^2 L + (L-2)n + 1) \right) = \mathcal{O}\left( n^2 L^2 \log(nL) \right).$$

For $n > L$, their bound becomes

$$\mathcal{O}\left(n^2 L^2 \log(n)\right);$$

for $L > n$, their bound becomes

$$\mathcal{O}\left(n^2 L^2 \log(L)\right).$$

It is worth noticing that when $r$ increases to $n$, our upper bound is consistent with the upper bound in (Bartlett et al., 2019). By (Bartlett et al., 2019), we can say the VC upper bound for the full-rank case is nearly tight. In Section 5, we will give a lower bound of a set of ReLU BRFCNN (not full rank case) and compare it with the upper bound in Theorem 3.

## 4 GENERALIZATION ERRORS FOR ReLU BRFCNNS

In this section, we first introduce the empirical error and the generalization error. Then we recall a well-known result, Theorem 4, in deep learning that related to VC dimension and these two errors. Finally, we derive Theorem 5, thus finding out how rank affects the generalization error in Remark 3.

**Definition 4** (Empirical Error and Generalization Error). *(Vapnik, 1968) Let $f$ be a function from the input space $\mathcal{X}$ to $\{-1, 1\}$. Let $D_m = \{(\boldsymbol{x}_1, y_1), \ldots, (\boldsymbol{x}_m, y_m)\}$ be the data set generated independently $m$ times from the distribution $\mathcal{D}$. The empirical error is defined by the mean zero-one loss on the data set $D$:*

$$\hat{E}(f; D_m) = \frac{1}{m} \sum_{i=1}^{m} \mathbb{I}(f(\boldsymbol{x}_i \neq y_i)),$$

*where $\mathbb{I}(\cdot)$ is a function that takes a boolean value as the input. If the input is $True$ it will return $1$ and $0$ otherwise. The generalization error is the expected zero-one loss of $f$ with respect to the data distribution $\mathcal{D}$:*

$$E(f; \mathcal{D}) = \mathbb{E}_{(\boldsymbol{x}, y) \sim \mathcal{D}}[\mathbb{I}(f(\boldsymbol{x} \neq y)].$$

**Theorem 4.** *(Vapnik, 1968) Let $\mathcal{F}$ be a given function class and $VCD(\mathcal{F}) = d$. The data set is $D_m$. Then we have for any $\delta > 0$, with probability at least $1 - \delta$, the following holds for $f \in \mathcal{F}$:*

$$\left| E(f; \mathcal{D}) - \hat{E}(f; D_m) \right| \leq \sqrt{\frac{8d \log(m/d) + 8 \log(\frac{4}{\delta})}{m}}.$$

Now we can derive Theorem 5.

**Theorem 5.** *Let $n > r$ and $0 \leq \delta \leq 1$. Let $\mathcal{F}$ be a ReLU BRFCNN with width at most $n$, depth at most $L$, and weight matrix rank at most $r$. Then for all $f \in F$, there exist a constant $C$ such that the following inequality holds:*

$$\mathbb{P}\left( \left| E(f; \mathcal{D}) - \hat{E}(f; D_m) \right| \leq C \sqrt{\frac{d}{m} log \frac{m}{d} - \frac{1}{m} log(\delta)} \right) \geq 1 - \delta.$$

*For $m > d$, we have*

$$\mathbb{P}\left( \left| E(f; \mathcal{D}) - \hat{E}(f; D_m) \right| \leq C \sqrt{\frac{d}{m} log(m) - \frac{1}{m} log(\delta)} \right) \geq 1 - \delta.$$

*Furthermore, for $n < L$, we have*

$$\mathbb{P}\left( \left| E(f; \mathcal{D}) - \hat{E}(f; D_m) \right| \leq C \sqrt{\frac{L^2 nr \log L}{m} log(m) - \frac{1}{m} log(\delta)} \right) \geq 1 - \delta.$$

*For $n \geq L$, we have*

$$\mathbb{P}\left( \left| E(f; \mathcal{D}) - \hat{E}(f; D_m) \right| \leq C \sqrt{\frac{L^2 nr \log n}{m} log(m) - \frac{1}{m} log(\delta)} \right) \geq 1 - \delta.$$

**Remark 3.** *Assume $L$, $n$ and $\delta \to 0$ are fixed in the setting of Theorem 5. And let the number of data points $m > d$. Then we get with very high probability,*

$$\left| E(f; \mathcal{D}) - \hat{E}(f; D_m) \right| \leq C\sqrt{r}\sqrt{\frac{log(m)}{m}}. \quad (C \text{ is a constant.}) \tag{4}$$

*According to Inequality (4), we get that the convergence rate of the generalization error is sensitive to the perturbation of $r$ when $r$ is small. On the contrary, when $r$ is large, the perturbation on it will not make the convergence rate of the generalization error change significantly, because the convergence rate of generalization error is roughly proportional to the $\sqrt{r}$ when we fix all other variables. Finally, although $r$ becomes very large we still can make $\hat{E}(f; \mathcal{D})$ converge to $E(f; D_m)$, since we can take infinitely many data points.*

## 5 LOWER BOUNDS OF VC DIMENSIONS FOR BRFCNNS

In this section, we construct a BRFCNN that can achieve the VC dimension close to the upper bounds we derived, which shows that our upper bound is nearly tight. Finally, we compare this bound with Theorem 3 in Remark 4.

**Theorem 6.** *Let $r, n \in \mathbb{Z}^+$ and $n > r \geq 6$, then there exists a ReLU BRFCNN with width at most $n$, $3 + 5\left(\lfloor \frac{r}{2} \rfloor - 2\right)$ layers, $\frac{41(\lfloor \frac{r}{2} \rfloor - 3)}{2} + \frac{15(\lfloor \frac{r}{2} \rfloor - 3)^2}{2}$ hidden nodes, and each weight matrix has rank at most $r$, such that its VC-dimension is at least*

$$\left(n - \left(\lfloor \frac{r}{2} \rfloor - 3\right)\right)\left(\lfloor \frac{r}{2} \rfloor - 3\right) = \Omega(nr).$$

*(See proof in Appendix A.10).*

**Remark 4.** *In order to compare the result of Theorem 6 with Theorem 3, we also want the depth of the network we constructing in Theorem 6 at most $L$. This enforces*

$$3 + 5m \leq L \implies m \leq \lfloor \frac{L-3}{5} \rfloor.$$

*Hence, for $\frac{L-3}{5} < \frac{r-6}{2}$, the lower bound becomes*

$$n'm = \left(n - \lfloor \frac{L-3}{5} \rfloor\right)\left(\lfloor \frac{L-3}{5} \rfloor\right) = \Omega(nL);$$

*for $\frac{L-3}{5} > \frac{r-6}{2}$, the VC lower bound will becomes*

$$n'm = \left(n - \left(\lfloor \frac{r}{2} \rfloor - 3\right)\right)\left(\lfloor \frac{r}{2} \rfloor - 3\right) = \Omega(nr).$$

*If $n >> L, r$, the lower bound we achieve here is close to the upper bound in Theorem 3, which shows that our upper bound is nearly tight.*

## 6 CONCLUSION AND FUTURE WORK

In this paper, We derive some upper bound for the VC dimension of BRFCNNs, and compare this upper bound for different settings. In addition, we also analyze how the ranks affect the generalization bounds. we also construct a BRFCNN that can achieve the VC dimension close to the upper bounds we derived, which shows that our upper bound is nearly tight. Finally, it is worth mentioning that the core method we propose in Section 3.1 can also be used to find a VC upper bound of a set of FCNNs that have orthogonal weight matrices. Notice that, the orthogonal matrices $M$ we consider here only satisfy the following property:

$$MM^T = D,$$

where $D$ is unnecessary to be an identity matrix. Since all the non-free variables can be expressed as rational fractions of free variables, we just need to follow the process in Section 3.4 again to get an upper bound.

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

## A APPENDIX

### A.1 PROOF OF THEOREM 1

*Proof.* Let the $m$ rational fractions be

$$f_1(x) = \frac{p_1(x)}{q_1(x)}, \cdots, f_m(x) = \frac{p_m(x)}{q_m(x)}.$$

Next, multiply all rational fractions by the square of their denominators

$$f'_1(x) = q_1^2(x) \times \frac{p_1(x)}{q_1(x)}, \cdots, f'_m(x) = q_m^2(x) \times \frac{p_m(x)}{q_m(x)},$$

$$\implies f'_1(x) = q_1(x)p_1(x), \cdots, f'_m(x) = q_m(x) \times q_m(x),$$

$$\implies K = |\{(\text{sgn}(f'_1(x)), \ldots, \text{sgn}(f'_m(x)) : x \in \mathbb{R}^n\}|.$$

Since all sign patterns(vectors) do not change their sign for multiplying positive numbers to their entries. Because

$$f'_1(x), \ldots, f'_m(x)$$

are polynomials function with degree at most $d_{den} + d_{num}$, by Theorem 1, we get $K \leq 2(2em(d_{den} + d_{num})/n)^n$. $\square$

### A.2 PROOF OF LEMMA 2

*Proof.* Since the rank of $M$ is $r$, we can find $r$ linear independent rows from $M$. The sub-matrix form by that $r$ rows $M_r$ has rank $r$, so we can extract $r$ columns from it such that these sub $r$ rows are still linear independent. Thus these $r$ columns form a $r \times r$ sub-matrix with rank $r$ and we denote this sub-matrix as $S_r$. The rest of the $m - r$ rows can be represented as a linear combination of that $r$ rows. In order to determine the linear combination coefficient, we only need to know the $r$ entries correspond to the columns of $S_r$ for the $m - r$ rows. More precisely we can let $M_i$ be the $i$ row of $M$ and it is also the rest of the $m - r$ rows, $v_i$ be a sub-vector of $M_i$ and share the same columns with $S_r$. By solving the following linear equation with respect to the linear combination coefficient $x_i$:

$$x_i S_r = v_i.$$

This linear equation has a unique solution, since $\boldsymbol{S}_r$ is a full-rank matrix. So, We can get the linear combination coefficient $\boldsymbol{x}_i$ of row $i$ by the $r$ independent rows of $\boldsymbol{M}$. Finally, we can recover row $i$ by

$$\boldsymbol{M}_i = \boldsymbol{x}_i \boldsymbol{M}_r.$$

Hence, we only need to know the $r$ linearly independence rows and $r$ entries for each of the $m - r$ rows to recover $\boldsymbol{M}$. We called these entries free variables and other entries of $\boldsymbol{M}$ called non-free variables. By simple calculation, there are $(n - r) \times (m - r)$ non-free variables and $(n + m - r) \times r$ free variables. $\qquad\square$

### A.3 PROOF OF LEMMA 3

*Proof.* By Lemma 2, we can find a $r \times r$ sub-matrix $\boldsymbol{S}_r$ with rank $r$ in matrix $\boldsymbol{M}$ and the rows and columns it corresponds to contain all the free variables of $\boldsymbol{M}$. Due to $r < min(m, n)$, $\boldsymbol{M}$ must contain non-free variables. Let $M_{i,j}$ be a non-free variable in $\boldsymbol{M}$ and we want to express it by only using the free variables. Let $\boldsymbol{S}_{ij}$ be a $(r + 1) \times (r + 1)$ matrix contains $\boldsymbol{S}_r$ and all the free variables in row $i$ and column $j$ and the element $M_{i,j}$. For simplicity, we denote matrix $S_{ij}$ as $\boldsymbol{S}'$ and $D_{ij}$ be the minor (the determinant of the sub-matrix of $\boldsymbol{S}'$ by removing the $i$th row and $j$ column) of the entry $S'_{ij}$. Then we have

$$rank(\boldsymbol{M}) = r,$$
$$\implies rank(\boldsymbol{S}') \leq r < r + 1,$$
$$\implies det(\boldsymbol{S}') = 0. \ (\boldsymbol{S}' \text{ is a } (r+1) \times (r+1) \text{ matrix}).$$

Let $S'_{ab}$ be the only non free variable in $S'$, by using the Laplace expansion along the $a$-th row we get

$$\sum_{j=1}^{r+1} (-1)^{a+j} S'_{aj} D_{aj} = 0,$$

$$\implies \sum_{j \neq b} (-1)^{a+j} S'_{aj} D_{aj} = (-1)^{a+b+1} S'_{ab} D_{ab},$$

$$\implies \frac{\sum_{j \neq b} (-1)^{a+j} S'_{aj} D_{aj}}{(-1)^{a+b+1} D_{ab}} = S'_{ab}. \tag{5}$$

According to the definition of determinant, all minors $D_{aj}$ and $D_{ab}$ are polynomials with degree $r$. In addition, $S'_{aj}$ are all free variables. Hence by equation (5), we get the numerator degree and denominator degree are $r + 1$ and $r$ respectively. $\qquad\square$

### A.4 PROOF OF LEMMA 4

*Proof.* Let $x_i$ and $y_i$ be the input and output of any unit in layer $i$ and $d_i$ be the maximum possible degree of the input polynomial $x_i$ with respect to the network parameters. According to the definition of FCNN, we know the input of an arbitrary input unit only depends on the input $x_0$, so $d_0 = 0$ which is the base case of $d_i$. If we know the input of any unit of layer $i$ is $x_i$, the output of this unit can be calculated as $y_i = \psi(x_i)$. The degree of the output polynomial is at most $dd_i$ since $\psi$ is a piecewise polynomial function of degree at most $d$. Because the input of layer $i + 1$ for arbitrary unit $j$ is defined as:

$$x_{i+1} = \sum_{k=1}^{k_i} y_k w_{k,j} + b_j.$$

$w_{k,j}$ is the weight associate with the unit $k$ in layer $i$ and the unit $j$ in layer $i + 1$. $b_j$ is the bias of unit $j$ in layer $i + 1$. Hence $d_{i+1} = dd_i + 1$ which is the recurrent relation of $d_i$. By using the base case and the recurrent relation for $d_i$, we get the following sequence:

$$d_0 = 0, d_1 = 1, d_2 = d + 1, d_3 = d^2 + d + 1, \cdots$$

From this pattern, we get

$$d_i = \sum_{t=0}^{i-1} d^t.$$

$\square$

### A.5 PROOF OF LEMMA 5

*Proof.* By the proof A.4 of Lemma 4, we know the polynomial degree of unit output is at most $d$ times than the polynomial degree of unit input. In order to arrive at layer $l$, $v_i$ needs to go through $l - i$ computation units. Hence $d^{l-i}$ is the maximum possible degree for $v_i$ can achieve.

$\square$

### A.6 PROOF OF LEMMA 6

*Proof.* When the weight matrices for each layer are not full rank we can replace all nonfree variables as free variables, the input of any unit in any layer $i$ is not a polynomial function of the free variables and biases. it is a sum of rational fractions since every non-free variable can be expressed as a ratio of free variables by Lemma 3. In order to write the sum of rational fractions as a single rational fraction(ratio of two polynomials) and determine the degrees, we need to find the common denominator of all the rational fractions of the sum. We know that only the non-free variables produce rational fractions, So one of the common denominators is $\prod_v v^{v_l}$. $v$ denotes any non-free variable form layer 1 to $l$ and all of them have been replaced by the free variables. $v_l$ be the maximum possible degree of $v$ for the input in layer $l$. The constructions of the common denominator are by assuming all the polynomials of the denominators for all $v$ are coprime, which means the common factor for these polynomials is 1. Hence, the common denominator is the product of all power of these polynomials. It is also worth noticing that the degree of this common denominator is an upper bound for the degree of the lowest common denominator which is used for the reduction of rational fractions to a common denominator. By Lemma 2, we know that there are $(k_{i-1} - r_i) \times (k_i - r_i)$ non-free variables in layer $i$. By Lemma 2 the polynomial degree of input of layer $l$ with respect to the non-free variables in layer $i$ is no more than $d^{l-i}$ by 5. By Lemma 3 the degree of the denominator of non-free variables after expressing by the free variables becomes $r_i$. The degree of the common denominator with respect to the free variable is at most

$$\sum_{i=1}^{l} (k_{i-1} - r_i) \times (k_i - r_i) \times d^{l-i} \times r_i. \tag{6}$$

After using the reduction of rational fractions to a common denominator for all rational fractions, we start to find which rational fraction have the maximum numerator degree. Let $v'$ denote free variables for weights, and $b$ denote the biases. $b_d, v'_d$ and $v_d$ mean the power of $b$, $v'$ and $v$. Finally, let $cd$ be the common denominator. We can write any rational fractions by

$$\prod_{v'} v'^{v'_d} \prod_v v^{v_d} \prod_b b^{b_d}.$$

Since $v'$ and $b$ are network parameters, So

$$\prod_{v'} v'^{v'_d} \text{ and } \prod_b b^{b_d}$$

only form the part of the numerator for each rational fraction, they contribute

$$\sum_{v'} v'_d + \sum_b b_d$$

degrees. Let $v_{num}$ and $v_{den}$ denote the numerator and the denominator of $v$. After the reduction of rational fractions to a common denominator $cd$. The degree of the numerator of $\prod_v v^{v_d}$ becomes

$$\sum_v deg(v_{num})v_d + cd - \sum_v deg(v_{den})v_d = \sum_v (deg(v_{num}) - deg(v_{den}))v_d + cd$$

$$= \sum_v v_d + cd.$$

by Lemma 3. Hence the degree of the numerator becomes

$$\sum_{v'} v'_d + \sum_b b_d + \sum_v v_d + cd.$$

The degree for the lowest common denominator $cd$ is upper bounded by the common denominator 6 and

$$\sum_{v'} v'_d + \sum_b b_d + \sum_v v_d \leq \sum_{t=0}^{l-1} d^t$$

by Lemma 4. We get the degree of numerator after the reduction of rational fractions to a common denominator is upper bounded by

$$\sum_{t=0}^{l-1} d^t + \sum_{i=1}^{l} (k_{i-1} - r_i) \times (k_i - r_i) \times d^{l-i} \times r_i.$$

$\square$

### A.7 PROOF OF THEOREM 3

*Proof.* By Lemma 2 and Theorem 2 and definition of $R$. We have

$$VCD(\mathcal{F}) = \mathcal{O}(L + (\sum_{l=1}^{L} \sum_{l'=1}^{l} k_{l'} + (k_{l'-1} + k_{l'} - r) \times r_{l'})$$

$$(\log(p) + \log(\sum_{l=1}^{L} k_l (\sum_{t=0}^{l-1} d^t + 2 \sum_{i=1}^{l} (k_{i-1} - r_i)(k_i - r_i)d^{l-i}r_i)))),$$

$$\implies VCD(\mathcal{F}) = \mathcal{O}(L + (\sum_{l=1}^{L} \sum_{l'=1}^{l} n + (2n - r) \times r)$$

$$(\log(p) + \log(\sum_{l=1}^{L} n(\sum_{t=0}^{l-1} d^t + 2 \sum_{i=1}^{l} (n-r)(n-r)d^{l-i}r)))).$$

By $d = 1$ and $p = 1$ for ReLU, we have

$$= \mathcal{O}(L + (\sum_{l=1}^{L} \sum_{l'=1}^{l} n + (2n - r) \times r)(\log(\sum_{l=1}^{L} n(l + 2 \sum_{i=1}^{l} (n-r)(n-r)r)))),$$

$$\implies VCD(\mathcal{F}) = \mathcal{O}\left(L^2 nr \log(n^3 r L^2)\right) = \mathcal{O}\left(L^2 nr \log(nrL)\right).$$

If the width of the network $n$ is larger than the depth $L$. We have

$$\mathcal{O}\left(L^2 nr\right) \mathcal{O}\left(\log(n^3 r L^2)\right) = \mathcal{O}\left(L^2 nr \log n\right).$$

If the depth $L$ of the network $n$ is larger than the width. We have

$$\mathcal{O}(L^2 nr)\mathcal{O}(\log(n^3 r L^2)) = \mathcal{O}\left(L^2 nr \log L\right).$$

$\square$

### A.8 PROOF OF UPPER BOUND COMPARISON

Let $U$ be sufficiently large. Let $A \times B = U$ and $A \geq B$.
For $L = B$ and $n = A$, there exist a constant $k_1 > 0$, such that

$$VCD(\mathcal{F}) = k_2 BUr \log(A);$$

For $L = A$ and $n = B$, there exist a constant $k_2 > 0$, such that

$$VCD(\mathcal{F}) = k_1 AUr \log(A).$$

By the assumption of $A$ and $B$. We get $A \geq \sqrt{U}$. Since $U$ is sufficient large we can say $A \geq \sqrt{U} > k_1, k_2$. Hence for a ReLU BRFCNN with a larger number of computational units and each layer share the share the same number of computational units, then when we swap the dimension of depth and width the VC upper bound will increase for $n < L$; the VC upper bound will decrease for $L > n$. Hence the network with the largest VC upper bound for $L = U, n = 1$ and the network with the smallest VC upper bound for $L = 2, n = U/2$. (VC upper bound 2 only works for $L \geq 2$).

A.9  FULLY CONNECTED BIT EXTRACTION NETWORK

Our construction of BRFCNN is motivated by (Bartlett et al., 2019), where they only consider full-rank wight matrix case. Let $S_n$ and $S_m$ denote the set of standard basis for $\mathbb{R}^n$ and $\mathbb{R}^m$. The class of bit extraction network is the set of functions

$$\mathcal{F}_b := \{f_b(x; (\boldsymbol{a} := a_1, \ldots, a_n)) : S_n \times S_m \to \{0, 1\},$$

$$\forall a_1, \ldots, a_n \in \{\frac{k}{2^m} : 0 \le k \le 2^m - 1\}.$$

There are $n$ parameters $a_1, \ldots, a_n$ for each of the network in $\mathcal{F}_b$. Let $(\boldsymbol{x}_1, \boldsymbol{x}_2)$ be the input vector and assume $\boldsymbol{x}_1 = \boldsymbol{e}_i$ and $\boldsymbol{x}_2 = \boldsymbol{e}_j$. The function $f_b$ define as

$$f_b(\boldsymbol{e}_i, \boldsymbol{e}_j; \boldsymbol{a}) := Bin(a_i)[j].$$

which means it returns the $j$ bits of the binary representation of the network parameter $a_i$. We can regard 0 as $-1$ and 1 as $+1$. By the definition of $\mathcal{F}_b$, for any sign patterns, we always can find a set of parameter $a$ such that $f_b(S_n \times S_m; \boldsymbol{a})$ generate this sign pattern. Hence, $S_n \times S_m$ can be shattered by $\mathcal{F}_b$ and $VCD(\mathcal{F}_b) = nm$. Next, we are going to introduce how to use ReLU FCNN to represent $\mathcal{F}_b$. Similar to the construction steps of (Bartlett et al., 2019), but adding many identity maps to ensure edges only link between adjacent layers. The identity mappings actually enforce the activation function to become the identity function. However according to the network input, all the inputs for all units are non-negative, hence we can only use the ReLU activation function in the Fully Connected Bit Extraction Network. The whole construction takes 3 step, the first step is by extracting the $a_i$ parameter from $\boldsymbol{a}$. The second step is extracting all bits from the binary representation of $a_i$ and arranging them into a size $m$ vector. The final step is returning the $j$-th bits of the vector in the second step. Before introducing the network in detail, let the vector $\boldsymbol{a}_{i,k}$ be the vector that contains the first $k$ bits of the binary representation of $\boldsymbol{a}_i$. The fist step also can view as a function take $(\boldsymbol{x}_i, \boldsymbol{x}_j)$ as inputs and return $(a_i, \boldsymbol{x}_j)$. The weight matrix for the first layer is defined as follows:

$$\begin{bmatrix} \boldsymbol{a} & \boldsymbol{0}_{1 \times m} \\ \boldsymbol{0}_{m \times n} & \mathbf{I}_{m \times m} \end{bmatrix}_{(m+1) \times (n+m)}.$$

The biases for all units are 0 for the first layer.

The second step takes $(a_i, \boldsymbol{x}_j)$ as input and return $(\boldsymbol{a}_{i,m}, \boldsymbol{x}_j)$. Let $q \in [m]$, This step will extract $q$ bits each time until all $m$ bits are extracted. We denote the layers for each $q$-bit extraction net as a bit extraction block. Then, the number of bit extraction blocks is $\lceil \frac{m}{q} \rceil$. This figure shows the first bit extraction block for $q = 3$. We use $b = b_1 b_2 b_3 \ldots b_m$ to denote the binary representation of $a_i$. This block takes size $m + 1$ vector $(b, \boldsymbol{x}_j)$ as input and return size $m + 4$ vector $(b_1, b_2, b_3, b_4 \ldots b_m, \boldsymbol{x}_j)$. The edges without number labeling mean the weight is 1. The 8 orange rectangle block means the indicator function with respect to the intervals. According to the output of these indicator functions, we can find out the location of $b$ in 3 decimals in interval $[0, 1)$. Hence we can get the first 3 bits from the network. After that, we just need to remove the first 3 bits in $b$. Which is done in the last layer of this bit extraction block.

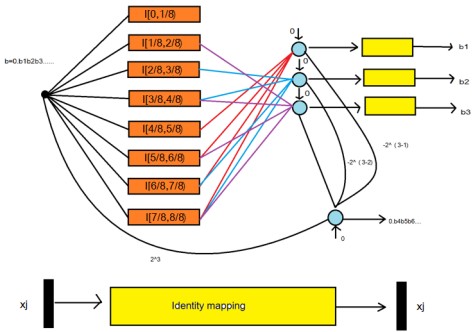

Figure 1: A diagram of a bit extraction block.

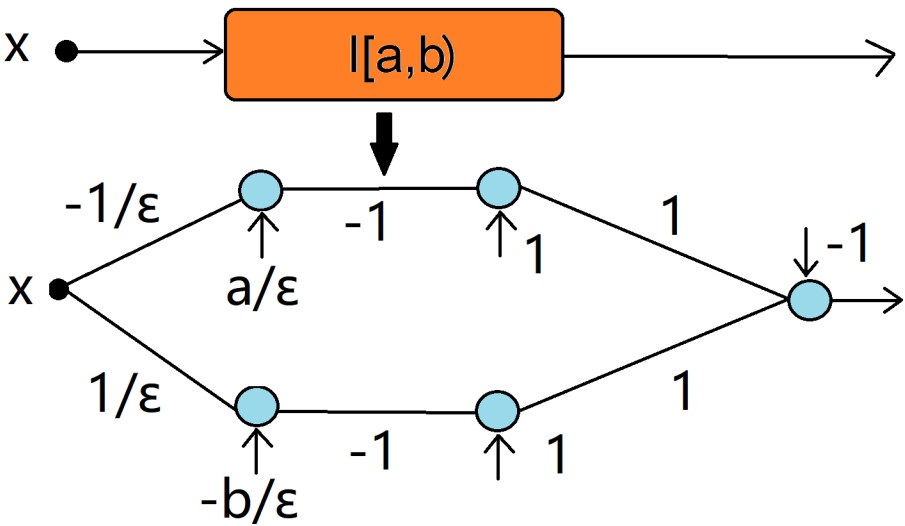

Figure 2: A diagram of an indicator net.

The indicator functions actually can be approximated by a sub-network. This subnetwork is defined by the function in (Bartlett et al., 2019):

$$f(x) := \sigma(1 - \sigma(\frac{a}{\epsilon} - \frac{x}{\epsilon})) + \sigma(1 - \sigma(\frac{x}{\epsilon} - \frac{b}{\epsilon})) - 1.$$

and When $\epsilon = 2^{-m-2}$, all the location of $a_i$ can be correctly identify. In conclusion, there are 5 layers for each bit extraction block.

The third step takes $(\boldsymbol{a}_{i,m}, \boldsymbol{x}_j)$ as input and return the $j$ element of $\boldsymbol{a}_{i,m}$. This can be implemented by 2 layers. The second last matrix is

$$[\mathbf{I}_{m \times m} \quad \mathbf{I}_{m \times m}]_{m \times 2m.}$$

The biases for all units are $-1$ for this layer. The last matrix is a column of 1s with size $m$ and bias $0$. According to the structure of the fully connected bit extraction network, the minimum depth is $8$.

### A.10   PROOF OF THEOREM 6

*Proof.* The structure of this ReLU BRFCNN actually is a Fully Connected Bit Extraction Network in Appendix A.9. The definition of $m$ and $q$ are defined in Appendix A.9. Let $n'$ be $n$ in Appendix A.9. The $n$ in Theorem 6 means the maximum number of units in each layer. To make the calculation easier, we assume that $m$ is divided by $q$ and $T = \frac{m}{q}$. According to the structure of the Bit Extraction FCNN, the following are all the sizes of the weight matrices and the corresponding inequalities that ensure all the ranks $\leq r$.
The first layer:
size:

$$(m + 1) \times (m + n').$$

inequality

$$m \leq r - 1 \text{ (Since } n' \geq 1).$$

Let $t \in [T]$, layer 1 of the bit extraction block $t$ has size:

$$\left(2^{q+1} + 1 + m + (t - 1)\right) q \times (1 + m + (t - 1)q),$$

and inequality

$$1 + m + (t - 1)q \leq r.$$

$$\implies m \le r - (t-1)q - 1.$$

The intersection of the $T$ inequalities become $m \le \frac{r+q-1}{2}$. This is also the inequality for $t = T$.

Layer 2 of the bit extraction block $t$

size:

$$\left(2^{q+1} + 1 + m + (t-1)q\right) \times \left(2^{q+1} + 1 + m + (t-1)q\right).$$

The intersection of the $T$ inequalities become $m \le \frac{r+q-2^{q+1}-1}{2}$.

Layer 3 of the bit extraction block $t$:

size:

$$\left(2^q + 1 + m + (t-1)q\right) \times \left(2^{q+1} + 1 + m + (t-1)q\right).$$

The intersection of the $T$ inequalities become $m \le \frac{r+q-2^q-1}{2}$.

Layer 4 of the bit extraction block $t$:

size:

$$(q + 1 + m + (t-1)q) \times (2^q + 1 + m + (t-1)q).$$

The intersection of the $T$ inequalities become $m \le \frac{r-1}{2}$.

Layer 5 of the bit extraction block $t$:

size:

$$(q + 1 + m + (t-1)q) \times (q + 1 + m + (t-1)q).$$

The intersection of the $T$ inequality become $m \le \frac{r-1}{2}$.

The second last layer:

$$m \times 2m; m \le r.$$

The last layer:

$$1 \times m; 1 \le r.$$

Finally, we need to find the intersection of the following inequalities

$$m \le r - 1, m \le \frac{r+q-1}{2}, m \le \frac{r+q-2^{q+1}-1}{2}, m \le \frac{r+q-2^q-1}{2}, m \le \frac{r-1}{2}, m \le r,$$

$$\implies m \le \frac{r+q-2^{q+1}-1}{2}$$

Let $q = 1$, $m = \lfloor \frac{r}{2} \rfloor - 3$ and $n' = n - (\lfloor \frac{r}{2} \rfloor - 3)$, then the VC dimension of the network will becomes:

$$n'm = (n - (\lfloor \frac{r}{2} \rfloor - 3))(\lfloor \frac{r}{2} \rfloor - 3) = \Omega(nr).$$

For fix $n'$, we also can calculate the number of nodes between the first layer and the second last layer(computational units) $U'$ by adding up the number of columns for all weight matrices from layer 1:

$$U' = 3m + \sum_{t=1}^{\frac{m}{q}} ((1 + m + (t-1)q) + \left(2^{q+1} + 1 + m + (t-1)q\right) +$$

$$\left(2^{q+1} + 1 + m + (t-1)q\right) + (2^q + 1 + m + (t-1)q) + (q + 1 + m + (t-1)q)).$$

For $q = 1$, we have

$$= 3m + \sum_{t=1}^{m} 5m + 5t + 11 = \frac{41m}{2} + \frac{15m^2}{2}.$$

For $m = \lfloor \frac{r}{2} \rfloor - 3$, we get

$$U' = \frac{41(\lfloor \frac{r}{2} \rfloor - 3)}{2} + \frac{15(\lfloor \frac{r}{2} \rfloor - 3)^2}{2}.$$

$\square$

