# OpenReview forum: "VC dimensions for deep neural networks with bounded-rank weight matrices"
_ICLR.cc/2024/Conference — Submitted to ICLR 2024_

### Official Review · Reviewer_1Lrg · 2023-10-30

**Soundness:** 3 good
**Presentation:** 2 fair
**Contribution:** 3 good
**Rating:** 6
**Confidence:** 3

**Summary:**

The paper gives an upper bound on the VC dimension of DNNs under the assumption of a bounded rank. This bound is then used to prove generalization bounds and finally a lower bound is given which matches the upper bound in some limits.

**Strengths:**

By assuming a rank of $r$, the dependence of the VC-dimension on the width $n$ of network changes from $n^2$ to $nr$ which can be a significant change if the rank is much smaller than the width. The tools used could be eneralized to other type of constraints as mentioned in the conclusion.

**Weaknesses:**

The readability of the paper could be improved, there are many Lemmas that are simply stated one after the other without any discussion.

Though the dependence of the bounds on the width are improved, they still fail to explain the empirical observation that the test error generally decreases with $n$ in practice.

**Questions:**

Some recent results (https://arxiv.org/abs/2305.19008) suggest that the (approximate) rank of the weight matrices could vary between layers with high rank layers towards the beginning and end of the network and approximately low rank weights in the middle. How would this affect the VC dimension?

It seems unlikely that the learned weights are exactly low rank, do you think that the bounds could be generalized to approximately low rank matrices?

---

### Official Review · Reviewer_E8PL · 2023-10-31

**Soundness:** 3 good
**Presentation:** 2 fair
**Contribution:** 2 fair
**Rating:** 3
**Confidence:** 2

**Summary:**

The paper establishes upper bounds for the VC dimension of BRFCNNs, analyzes the influence of different ranks on generalization bounds, and demonstrates a BRFCNN with a VC dimension close to the upper bounds, suggesting a nearly tight upper bound and insights into network generalization capacity.

**Strengths:**

1. The paper derives upper bounds for the VC dimension of BRFCNNs and compares these upper bounds for various settings.

2. The analysis explores how different ranks impact the generalization bounds of BRFCNNs.

3. The paper presents a constructed BRFCNN that achieves a VC dimension close to the upper bounds derived. This demonstration indicates that the upper bound is nearly tight, providing valuable insights into the network's capacity for generalization.

**Weaknesses:**

I am not familar with the topic of VC dimensions for DNNs, and I cannot fully understand the technical contribution of this paper. As a result, I am unable to provide constructive review comments for this paper. However, I strongly advise the authors to revise the paper's main body and offer more in-depth discussions regarding the insights behind the technical contribution.

**Questions:**

Could you please clarify the technical challenge in this topic? It would help me understand the technical contribution.

---

### Official Review · Reviewer_YfBS · 2023-11-02

**Soundness:** 2 fair
**Presentation:** 1 poor
**Contribution:** 2 fair
**Rating:** 3
**Confidence:** 3

**Summary:**

This paper focuses on driving a VC dimension of a fully connected feedforward network, with peace-wise polynomial activations and bounded rank fully connected matrices. The authors main theoretical contributions is showing that the VC dimension is bounded by $O( L^2 n r \log (p n d)$ where $n$ denotes constant width of layers, $r$ is rank of fully connected layers, $p$ is the maximum number of intervals int he piece-wise activations, $d$ is the maximum degree of these polynomials, and $L$ is the network depth. The paper further argues that this upper bound on VC dimension is almost tight for a ReLU network, since it reaches a VC dimension of $O(L^2 nr \log(n). $
By plugging their result into the classic VC dimension, generalization error bound by Vapnik, they arrive at a generalization bound for the bounded rank fully connected networks.

**Strengths:**

The authors pick a seemingly novel problem, the VC dimension of a rank-bounded fully connected networks. The main result on VC dimension implies a generalization bound for these function classes. In the particular case of ReLU network, the error bound roughly grows as $\sqrt{L^2 n r / m}$ which is in meany ways interesting. While a direct test of this result may not be very feasible, it is a pity that there is no empirical evidence is presented to see if this bound is valid.

**Weaknesses:**

Main critique: My big issue with the paper is that the ideas are not presented well, and the technical/notation issues impede the understanding of the reader. While reading the paper I had to frequently go back and forth to try and find the definition of a variable. In other places, there were some parts were very simple concepts (like how activations and fully connected layers are defined), have are explained in an unnecessarily complicated and twisted ways. I've tried to aid the authors with  improving these presentation issues in my detailed issues/questions list.

Main technical/issues questions:
- "Due to the rank constraint for the weight matrices, $f (x_j ; a)$ are not necessarily polynomials, instead they could be rational fractions." This sentence is not elaborated on, and on to the best of my understanding, is not correct. a low-rank matrix $W$ can be written as $W = AB$, and any polynomial activation $\psi$ applied on the matrix $\psi(W x),$ where $x$ denotes the previous layer, is a polynomial in the space of the inputs. Please correct me if I'm wrong in making these hand wavy conclusions.
- Remark 3: what happened to the rest of the bound, namely $L^2 n r \log(n)$? Please correct me if I'm wrong, but from what I can see, even if we ignore the $-\log\delta$-term, the first part is suggesting the generalization error is bounded by $C \sqrt{L^2 n r \log(n)\log m / m}$ while the remark 3 is suggesting only $C \sqrt{d}\sqrt{\log m/ m}$  holds with high probability, which is substantially smaller.
- Remark 3: Formally speaking, if $\delta$ converges to $0$ then the failure probability term will converge to infinity $-\log(\delta)\to \infty$. Taken from wikipedia, with high probability means "an event ... whose probability depends on a certain number n and goes to 1 as n goes to infinity," I am guessing that what authors want to say is that for some arbitrarily small of $\delta,$, then we can adjust the value $C$ such that the bound will hold with a probability that will go to zero with increasing $m$. For example if we set $\delta = m^ {-L^2 n r\log(n)}$, then the bound for $\sqrt{2}C$ will hold with $1-\delta$ probability, which will converge to $0$ as $m$ increases.  Again, please correct me if I'm wrong.

Detailed issues/questions:
- p2 , end of page, let $x_1, \dots, x_n\in \mathcal{F}$ this seems to be a typo, shouldn't they belong to $X$ instead?
- page 3 mentions $w = (w_1,\dots, w_W)\in R^W$ and bias $b = (b_1,\dots, b_U)\in R^U. $ from what I see, I don't see the definition of $W$ and $U$ anywhere in the text. Also, the definition of weights seems to be a matrix, while here it looks like a vector of dimension $W$. I think it's better to clarify this notation a bit and use a simpler, more standard notation
- page 3, while strictly speaking this is not a problem, the variable z as the pre-activation variable, $\Psi_{i+1}(z) = (\psi(z_1),\dots, \psi(z_{k_{i}+1})$ is a confusing at first read. Define $z$ as pre-activation first?
- page 5, Thm2: "Let wi be the number of network parameters from layer 1 to layer i." I find this sentence quite confusing and hard to understand. what do you mean here?
- page 8 Thm5: there seems to be a conflict of variable $d$ which was previously defined as maximum degree of the polynomial, but here it is used as the bound on the VC dimension. This is particularly troublesome because the first two equations in Thm5 invoke the Thm2 result which does have a "d", and use it in Thm4 which also has d but with a different meaning. Perhaps a change of name of variables?

**Questions:**

- I wonder what is the main motivation for this theoretical case study? What is the surprising conclusion of this result which we can draw and apply on some problem? From what I can see this is somewhat missing from the current manuscript. While this is not a direct criticism, including such an explanation will go a long way to persuade and engage the future readers.
- If I understand correctly the result on VC dimension of fully connected network has been addressed previously, and the main result here is about the case where fully connected layers are low-rank. Without delving into too much technical depth, the fact that the degrees of freedom in a rank-limited fully connected weight matrix will reduce from $n^2$ to $n r$ seems like a borderline obvious thing. Please convince me otherwise, that this is not a simple result?

---

### Official Review · Reviewer_12gB · 2023-11-03

**Soundness:** 2 fair
**Presentation:** 2 fair
**Contribution:** 2 fair
**Rating:** 3
**Confidence:** 4

**Summary:**

The paper explores the VC dimensions of DNNs with piecewise polynomial activations and bounded-rank weight matrices. It presents an upper bound for these VC dimensions, constructs a ReLU DNN that reaches this bound, and compares the generalization power of various DNN architectures.

**Strengths:**

The paper studies an important problem. The comparison of different DNN architectures in terms of their VC dimensions could be a useful tool in the future.

**Weaknesses:**

The main weakness of the paper is the claim of novelty. The authors assert that this is the first study of the VC dimensions of low-rank DNNs, which is contradicted by the existence of prior work [1]. This oversight undermines the originality of the paper. The authors should have acknowledged this previous work and clearly delineated their contributions relative to it.

Another potential weakness is the lack of empirical VC dimensions. In [1], the authors reported empirical VC dimensions, which provided a practical validation of their theoretical results. The inclusion of similar empirical results would strengthen the current paper.

[1] Empirical Study on the Effective VC Dimension of Low-rank Neural Networks, ICML 2021 OPPO

**Questions:**

- How do the authors differentiate their work from the prior study [1] on the VC dimensions of low-rank DNNs? A clear comparison of the two studies would be beneficial.
- Can the authors provide empirical VC dimensions to validate their theoretical results, as was done in [1]? This would enhance the practical relevance of the paper.

---

### Meta-Review · Area_Chair_7NyL · 2023-12-12

**Metareview:**

Summary: The article studies the VC dimension of networks with piecewise polynomial activation and bounded rank weight matrices.

Strengths: Reviewers find the paper studies an important, seemingly novel problem.

Weaknesses: Main concerns are the presentation and lack of discussion.

I find that although the article proposes interesting results, I agree that the presentation and discussion, particularly of implications, need improvements. Reviewers have provided diverse suggestions. The authors did not respond to the initial reviews. Hence I am recommending to reject the article.

**Justification For Why Not Higher Score:**

The authors did not respond to the initial reviews.

**Justification For Why Not Lower Score:**

NA

---

### Decision · Program_Chairs · 2024-01-16

Reject